# Structured Long-Chain Omega-3 Fatty Acids for Improvement of Cognitive Function during Aging

**DOI:** 10.3390/ijms23073472

**Published:** 2022-03-23

**Authors:** Ignasi Mora, Lluís Arola, Antoni Caimari, Xavier Escoté, Francesc Puiggròs

**Affiliations:** 1Brudy Technology S.L., 08006 Barcelona, Spain; 2Nutrigenomics Research Group, Departament de Bioquímica i Biotecnologia, Universitat Rovira i Virgili, 43007 Tarragona, Spain; 3Eurecat, Centre Tecnològic de Catalunya, Biotechnology Area, 43204 Reus, Spain; antoni.caimari@eurecat.org (A.C.); francesc.puiggros@eurecat.org (F.P.); 4Eurecat, Centre Tecnològic de Catalunya, Nutrition and Health Unit, 43204 Reus, Spain; xavier.escote@eurecat.org

**Keywords:** structured lipids, omega-3 PUFAs, cognitive function, cell senescence, DHA, EPA, omic technologies

## Abstract

Although the human lifespan has increased in the past century owing to advances in medicine and lifestyle, the human healthspan has not kept up the same pace, especially in brain aging. Consequently, the role of preventive health interventions has become a crucial strategy, in particular, the identification of nutritional compounds that could alleviate the deleterious effects of aging. Among nutrients to cope with aging in special cognitive decline, the long-chain omega-3 polyunsaturated fatty acids (ω-3 LCPUFAs) docosahexaenoic acid (DHA) and eicosapentaenoic acid (EPA), have emerged as very promising ones. Due to their neuroinflammatory resolving effects, an increased status of DHA and EPA in the elderly has been linked to better cognitive function and a lower risk of dementia. However, the results from clinical studies do not show consistent evidence and intake recommendations for old adults are lacking. Recently, supplementation with structured forms of EPA and DHA, which can be derived natural forms or targeted structures, have proven enhanced bioavailability and powerful benefits. This review summarizes present and future perspectives of new structures of ω-3 LCPUFAs and the role of “omic” technologies combined with the use of high-throughput in vivo models to shed light on the relationships and underlying mechanisms between ω-3 LCPUFAs and healthy aging.

## 1. Introduction

After the last hundred years of medical and life science technology progress and advances, people now live longer than ever before. Currently both preventive and therapeutic approaches are failing to reduce non-communicable diseases (NCDs), but they have succeeded in increasing our life-expectancy. Consequently, although human lifespan has significantly increased, our healthspan has not kept up the pace [1].

Demographically, society is getting older and the aging process includes progressive and irreversible biological changes, resulting in a growing risk of suffering from chronic diseases, cognitive impairments, physical disfunctions, and an increased probability of dying [1]. In fact, the loss of cognitive function is considered the most critical change during aging and it is projected that the patients with dementia—considered as significant loss of cognitive function which should be distinguished from neurodevelopmental disorders, such as intellectual disability [2]—will reach up to 115.4 million in 2050 [3].

The aging process is conditioned by the interactions between our genetic inheritance and environmental influences. While aging, our cells are submitted to a wide range of intrinsic and extrinsic insults, including oncogenic activation, oxidative and genotoxic stress, mitochondrial dysfunction, irradiation, and mutagenic agents [4]. In response to these disturbances, a stable state of cell cycle arrest happens and the cellular ability to proliferate decreases entering into a phase of senescence [4].

Senescent cells undergo morphology changes, chromatin remodelling, metabolic reprogramming and secrete a complex mix of mostly proinflammatory factors, like IL-1 and TNF-α. As senescent cells are more abundant, it leads to a potentially chronic inflammatory state independent from the activation of immune cells, which may impair tissue homeostasis [5]. This phenomenon of chronic low-grade systemic inflammation is called “inflammaging” and is considered to play a central role in the pace of aging, in the impairment of cognitive and physical functions and lastly, in the development of age-related disease.

In addition, many intrinsic and environmental factors generate oxidative stress which is a common phenomenon caused by an imbalance between production and detoxification of free radicals. Free radicals, mainly reactive oxygen and nitrogen species (RONS), can damage cells and tissues leading to the activation of proinflammatory pathways which contribute to the above mentioned “inflammaging” [6] leading to a higher degree of cellular senescence.

To improve the healthspan and the quality of life in the elderly, it is crucial to consider the role of preventive health interventions. Preventing disease, not only has positive health and well-being outcomes, which is the most important impact, but also wider economic significance, since the healthcare system is not prepared to handle the pressure of an aged society. Consequently, the goal of the scientific community is to find non-pharmacological therapies to prevent the most common age-related disfunctions and especially, those related with the loss of cognitive function, thus extending the well-being and optimal health of aging people for the longest possible time [7].

Following this approach, two powerful and recent strategies, functional foods and exercise, have been shown to decrease the risk of aging-related diseases. As nutrition is closely linked with health status, there is a growing demand for appropriate dietary patterns that include food supplements and functional foods to address healthy aging. Bioactive compounds with proven anti-inflammatory and antioxidant effects are suitable as anti-aging ingredients, but few of them like vitamin E, vitamin B12 and B6 or tea polyphenols have shown consistently improved cognitive effects [8,9,10,11].

Among nutrients assessed for brain health, the omega-3 polyunsaturated fatty acids (ω-3 PUFAs) must be highlighted, specifically the ω-3 LCPUFAs: DHA and EPA. Increased intake of ω-3 LCPUFAs, which are mainly found in fish and other seafood, have been associated with better cognitive function, slower rates of cognitive decline and an overall lower risk of developing dementia [12,13]. Furthermore, DHA and EPA are promising bioactive ingredients in the treatment of more severe neurological age-related disease like Alzheimer’s or Parkinson’s disease [14,15]. However, some clinical studies with healthy subjects have failed to prove a direct effect between cognitive function improvement and ω-3 LCPUFAs supplementation during aging [15].

Thanks to the research of bioactive compounds for healthy aging, new structured forms of ω-3 LCPUFAs have emerged as promising ingredients with more powerful effects than simpler forms of DHA and EPA. This group of structured PUFAs comprises a wide range of larger structures, like triglycerides, phospholipids or derived lipid mediators of ω-3 LCPUFAs, and they have been proven to have greater bioavailability and stronger anti-inflammatory and antioxidant effect than conventional ω-3 PUFAs [16,17] according to the findings from in vitro and preclinical studies.

This review summarizes the current perspectives of ω-3 LCPUFAs on the improvement of cognitive function during aging and compares their potential with new structures of ω-3 LCPUFAs. In addition, high-throughput technologies and new biological models for elucidating unknown mechanisms and as screening platforms for novel structures ω-3 LCPUFAs will be explored.

## 2. Senescence, Aging and Bioactive Compounds

When the phenomenon of senescence starts to grow in the nervous system, brain function impairment occurs. To understand the relationship between senescence onset and its negative implications on the nervous system and brain function, and how nutrition can modulate this relationship, the biological mechanisms of senescence should be updated.

### 2.1. Nutrients and Anti-Cellular Senescence Targets

As stated above, senescence is a physiological stress response of mammalian cells that results in the development of senescent cells (SC) with distinct physical, molecular, and metabolic signatures. Based on the type of induction, three broad categories of cellular senescence are defined: replicative senescence (telomere attrition), oncogene-induced senescence (activation of oncogenes), and genotoxic or oxidative stress-induced senescence.

Regardless of the different triggers of cellular senescence, SC are invariably accompanied by impaired mitochondrial functions, increased intracellular RONS and production and activation of the DNA damage response. Late SC, exhibit characteristic secretion of a milieu of cytokines and growth factors causing a chronic inflammation state independent from the activation of immune cells which may impair tissue homeostasis, a phenomenon known as inflammaging as mentioned above [18]. As a strategy to fight against tissue homeostasis impairment, three nutrition-mediated anti-cellular senescence targets may be established, according to the current literature.

#### 2.1.1. Redox Homeostasis

Increased oxidative stress can directly accelerate the development of cellular stress and establish a senescence program in different cells. Nutritional status can be of particular importance in this context as several studies have shown that consumption of antioxidant compounds like the green tea component epigallocatechin gallate (EGCG) [19], resveratrol [20], quercetin [21], vitamin C [22], and minerals such as zinc [23], have proven anti-cellular senescence attributes through improved redox homeostasis in both in vitro and in vivo studies [24].

#### 2.1.2. Cell-Cycle Regulation

The prolonged persistence of SC in tissues is a consequence of their ability to resist the triggering of apoptotic cell death due to deregulation of anti-apoptotic and pro-apoptotic pathways. It has been demonstrated that targeted apoptosis of SC can attenuate age-related dysfunctions, therefore applying strategies that can selectively remove SC are of considerable interest in reducing SC burden with age [25]. Natural compounds, like resveratrol [26], EGCG [27] and probiotic bacterial metabolites [28], have been identified as “senolytics” or “senomorphics” which can specifically ablate SC by inhibiting anti-apoptotic pathways (i.e., Bcl-2) or alter their phenotype by interfering with inflammatory pathways controlling deleterious paracrine effects of cellular senescence [18]. Emerging early clinical trials are showing promising results of senolytics and senomporhics such as improvments in age-related morbidity and mortality [29].

#### 2.1.3. Nutrient Sensors

Various evolutionarily conserved nutrient signalling pathways are strongly implicated in organismal aging and longevity [30]. Examples of these nutrient-sensing longevity pathways include the mechanistic target of rapamycin (mTOR) pathway, the sirtuin enzyme family, and the insulin/IGF-1 signalling pathway. Various bioactive substances such as resveratrol [31,32], berberine [33,34], curcumin [35,36], and probiotic bacterial metabolites [28], have shown the dual capacity to activate or inhibit these pathways. Although the underlying relationship between these nutrient-sensing pathways and aging are yet unclear; it has been established that these pathways are impaired by aging and their molecular targeting can positively increase healthspan and lifespan.

However, the specific reason behind the age-related accumulation of SC is not yet clear and seems to be related to the immune system. As age increases, the immune system undergoes characteristic age-related dysfunctions collectively called immunosenescence, which may compromise the ability of immune cells to effectively clear SC. Consequently, targeting immune cells for promoting efficient clearance of SC is rapidly emerging as a novel anti-aging strategy [18,37].

### 2.2. Omega-3 LCPUFAs to Cope with Senescence

Blood levels of EPA and DHA are in the low range for most of the world population [38] despite them being promising bioactive nutrients contributing to healthy aging. Both ω-3 LCPUFAs modify cellular function through overlapping and distinct mechanisms of action. An increased intake of EPA and DHA exerts an enhanced appearance of those fatty acids in the membrane phospholipids of cells, and they potentially cope with cellular senescence through the mechanisms outlined below [15,39].

#### 2.2.1. Specialized Pro-Resolving Mediators and Anti-Inflammatory Effects

Many of the biological actions of ω-3 LCPUFAs are driven via derived-lipid mediators. These bioactive lipids are a varied group of enzyme-derived oxygenated forms mainly with anti-inflammatory roles, of which the specialized pro-resolving mediators (SPMs) must be highlighted. The SPMs derived from the C20 PUFAs, such as EPA, are called eicosanoids, whereas those that come from C22 PUFAs, as DHA, are called docosanoids. Biosynthesis of these SPMs are catalysed by three enzyme systems: cyclooxygenase (COX), lipoxygenase (LOX), and cytochrome P450 (CYP450), and their biological effect its triggered via a series of different cell-type specific receptors. EPA produces E-series resolvins (RvE), whereas DHA produces protectins, D-series resolvins (RvD) and maresins (MaR) [14,40]. Some of these SPMs were proven to reduce chronic inflammation states characteristic of elderly in preclinical and clinical studies [40,41] through their inflammatory revolving effects acting as senomorphic agents [42,43,44].

In addition to the effect of SPMs, DHA and EPA also produce anti-inflammatory effects by changing the cell membrane composition. On one hand, they partially replace the ω-6 PUFAs of the membrane which are precursors of pro-inflammatory lipid mediators, specifically arachidonic acid (ARA), thus reducing pro-inflammatory substrates and allowing less systemic inflammation risk [45]. On the other hand, the ω-3 LCPUFAs make the cell membranes more fluid by replacing saturated fatty acids, which affects the behaviour of several membrane proteins and signalling platforms (lipid rafts) [46]. As a consequence, the transmission of inflammatory signals within cells becomes blunted, resulting in a reduced activation of proinflammatory transcription factors like nuclear factor kappa-light-chain-enhancer of activated B cells (NFκB) [47], and probably also showing a senomorphic effect.

Furthermore, DHA and EPA can be released from the membrane and transported to the nucleus where they can modify the gene expression through interaction with transcription factors, such as peroxisome proliferator activated receptors (PPARs) [47], thus modulating fat metabolism and anti-inflammatory pathways and theoretically affecting nutrient-sensing longevity pathways.

#### 2.2.2. Antioxidant Pathway

Omega-3 LCPUFAs also have a well described antioxidant capacity which could alleviate oxidative stress. Interestingly, despite ω-3 LCPUFAs being very prone to oxidation by accepting electrons and becoming oxidized lipids that might be harmful for the cells, they have the ability—particularly DHA—to enhance the activity of the antioxidant enzymes, improving detoxification and counteracting oxidative stress [48,49].

This phenomenon happens, hypothetically, because an increment of DHA in the cellular membrane phospholipids slightly increases ROS production through derived oxidated forms of ω-3 LCPUFAs, which induces antioxidant signalling and prepares the cell to be more resilient against oxidative stress. In fact, like a hormetic phenomenon, ω-3 LCPUFAs induces damage in a primary stage, but once the moderate oxidative stress is resolved, adaptation occurs [17,50].

DHA is able to activate antioxidant defences and prepares the cell to counteract future ROS threats through several pathways: (1) by regulating the nuclear factor erythroid 2 like 2 (NFE2L2) and its downstream target protein, heme-oxygenase-1 (HO-1), that controls the gene expression of a large variety of antioxidant, cytoprotective and detoxifying enzymes [17,51]; (2) by enhancing the activity of antioxidant enzymes like superoxide dismutase (SOD), catalase (CAT) [50,52], ɣ-glutamyl-cysteinyl ligase and glutathione reductase (GR) [53], and their synthesis through activation of transcription factors such PPARα [50] or NFE2L2 [52]; and (3) by significantly increasing reduced glutathione production (GSH), which decreases oxidative damage in cellular stress models, with or without stimulation of glutathione metabolism enzymes [51,52,53]. Quite detailed experimental studies have provided some lines of evidence of these interrelationships, however, the exact mechanism has still to be elucidated.

## 3. Bioactive Compounds and Improvement in Cognitive Function

A progressive decline in memory, language, problem-solving and other cognitive skills that affects a person’s ability to perform everyday activities, leads to mild cognitive impairment and may progress to dementia [54].

As mentioned previously, late senescent cells and changes in microglial function (immunosenescence) cause neuroinflammation and oxidative stress due to an enhanced pro-inflammatory cytokine production and weak redox homeostasis. Inflammation and ROS drive a progressive impairment of brain cell processes, such as neural membrane fluidity reduction, less synaptic plasticity and low neurogenesis [55]. These impaired functions may lead to irreversible neural changes like loss of grey and white matter volume, and significant alterations in memory, learning abilities and spatial recognition which have been described in both humans and animals [55,56] driving the age-related cognitive decline.

Although there is much to be clarified about the specific molecular mechanisms in which dietary components influence cognitive function, a growing literature supports the idea that certain dietary patterns and some bioactive compounds are able to modulate brain structure and function, exerting their beneficial influence throughout the entire lifespan [10,57].

Vitamins of the B group have been studied for their potential effect on cognitive function because of their role in homocysteine metabolism, specially vitamins B6 (pyridoxine), B9 (folate) and B12 (cobalamin) [11]. Several clinical studies have found that raised concentrations of homocysteine in plasma might be associated with increased risk of dementia in people older than 65 years [58]. Supplementation, over 3 years, of 0.8 mg a day of folate—higher than twice the recommended daily intake [59]—improved cognition in participants aged 50–70 years, but the intervention was more effective in those with high baseline homocysteine concentrations [60]. In general, most clinical trials of B vitamins have found no association with cognitive function. Only individuals with high baseline homocysteine, low baseline vitamin B concentrations, or established cardiovascular and cerebrovascular disease may benefit most from vitamin B supplementation [10].

As the brain is highly susceptible to oxidative damage, it has been suggested that inadequate antioxidant defences might mediate the pathogenesis and progression of dementia [61]. Many antioxidant nutrients such as vitamin C, vitamin E [62], zinc [63] (cofactor for enzymes with antioxidative activity) and carotenoids [64], and non-nutrient food ingredients like polyphenols [65], anthocyanins [66], lignans [67] or allicin [68] have been proven to be beneficial for age-related cognitive impairment. But, as far as there is not a deficit of the mentioned nutrients or a pathological situation, clinical trials with supplementation of these mentioned antioxidant compounds have not demonstrated a beneficial effect on any cognitive outcome in healthy patients [10].

Despite this, among nutrients addressed for brain health, ω-3 PUFAs, specifically ω-3 LCPUFAs must be highlighted [9]. Aging is associated with decreased cerebral ω-3 LCPUFAs levels due to reduced absorption, lower ω-3 PUFA capacity to cross the blood-brain barrier [12], and decreased capacity to convert PUFAs into LCPUFAs in the brain. As a major neuronal membrane component, DHA regulates neurogenesis, synaptogenesis, and neural membrane fluidity, which in turn modulates the speed of cell signalling and neurotransmission. In comparison with DHA, EPA constitutes a minimal proportion of total brain LCPUFA, but EPA inhibits proinflammatory metabolism and promotes adequate cerebral blood flow [69]. Altogether, a large amount of evidence demonstrates that poor ω-3 LCPUFA status in brain and plasma is associated with age-related cognitive decline [70].

## 4. Omega-3 LCPUFAs against Age-Related Cognitive Impairment

Although DHA has a structural role as a major component neuronal membrane fatty acid, it is endogenously transformed in the nervous system to an endocannabinoid-like metabolite called N-Docosahexaenoylethanolamine (synaptamide). Mainly through the specific target receptor GPR110, a G-protein coupled receptor, synaptamide promotes neurogenesis, neurite outgrowth and synaptogenesis in developing neurons. GPR110 induces cAMP production and phosphorylation of protein kinase A (PKA) and the cAMP response element binding protein (CREB) [71]. This signalling pathway leads to the expression of neurogenic and synaptogenic genes and suppresses the expression of proinflammatory genes. GPR110 is highly expressed in the brain during development but also during adulthood, emphasizing its relevance in the hippocampal region where neurogenesis is still happening [72,73].

Once it was proven that the pharmacological inhibition of neuroinflammation improved memory in aged murine models [74], the anti-inflammatory role of ω-3 LCPUFA gained attention as a possible therapeutic nutrient for healthy aging. Recent studies with aged rats [75] and mice [55] from a preventive point of view, show an improvement in memory and cognitive skills after a reduction in microglial activation and neuroinflammation, following supplementation with EPA and/or DHA. Moreover, in animal models ω-3 LCPUFAs and SPMs also have proven neuroinflammatory resolving effects and cognition improvements in age-related diseases, like Parkinson’s [40,76].

In preclinical data, human trials show a negative correlation between increased dietary supply of ω-3 LCPUFAs and pro-inflammatory markers [41] but, a positive correlation with verbal performance and learning ability in elderly people with high risk of early cognitive decline [40]. In patients with coronary artery disease and potentially ischemic risk, which reduces cerebral blood flow and contributes to the development of dementia, a daily high-dose of 1.86 g of EPA and 1.5 g of DHA over a 30 months period enhanced cognitive function significantly in comparison with control [77]. Also in middle-aged to older adults with obesity, which accelerates cognitive decline by endothelial dysfunction (impaired vasodilatation) in the peripheral and cerebral vasculature, greater processing speed mediated by improvements in circulatory function was observed following fish-oil supplementation (total dose of 0.4 g EPA and 2 g DHA) [78]. Furthermore, a study with old adults, with no risk factors, who had self-perceived cognitive function impairment (bad memory, learning difficulty,…), reported lower levels of cognitive inefficiency in activities of everyday life following 24 weeks treatment with fish oil (daily dose of 1.6 g EPA and 0.8 g DHA) [79].

However, when the effect of ω-3 LCPUFAs are assayed in healthy older adults the results are controversial [69]. Despite reported meta-analyses highlighting the potential of ω-3 PUFA to improve memory [13] and cognitive decline [70], some interventional studies have changed the perspective of the actual benefits of ω-3 LCPUFA [80,81]. As an example, Van de Rest et al. did not find significant changes in any of the cognitive domains for either low-dose (0.4 g/day EPA-DHA) or high-dose (1.8 g/day EPA-DHA) fish oil supplementation groups compared with placebo in a cohort of healthy individuals aged 65 years or older [82]. More recently, Baleztena et al. reported that daily supplementation with 0.75 g of DHA and 0.12 g of EPA did not show an improvement in the global cognitive function in adults over 75 years of age. They just found an apparent improvement in memory loss when the study subjects were well nourished [83].

According to the literature, two main issues justify the divergences [15]. Firstly, the effect of DHA and EPA depends on the stage of cognitive health assessed. This is supported by Canhada et al. in a recent systematic review where they conclude that the most beneficial effect of EPA and DHA supplementation in Alzheimer’s patients can only be expected in the early stage of the disease [84]. Secondly, there were several weaknesses in the trial protocols designed to assess the effect of EPA and DHA, such as the variability of doses of DHA and EPA, the type of placebo used, the combination of treatments [85], the duration of treatment, the sample size, the ω-3 LCPUFA status of the participants and the cognitive outcomes/tests measured as primary and secondary variables [15].

Therefore, since clinical trials are performed with a wide range of target population and low consistency of biomarkers measured in the nutritional interventions, the results from clinical studies with ω-3 LCPUFA do not allow a consensus about how traditional forms of ω-3 LCPUFAs affect cognitive function and aging [86].

## 5. Structured Lipids: Innovative Omega-3 LCPUFAs Molecules for Healthy Aging

Structured lipids can be defined as chemically or enzymatically modified lipids that change the fatty acid composition and/or the positional distribution [87]. They are created to be applied in functional food and clinical nutrition because of their characteristics or bioactive properties [88,89]. These structures can be a replica of natural forms of lipids with special structures designed to have a specific function. Most of the studies mentioned above use ethyl ester (EE) forms of DHA and EPA, or oils rich in DHA and EPA-EE for supplementation. However, in natural matrixes containing DHA and EPA, like in blue fish or breastmilk, they can be found in more complex structures as phospholipids (PL) or triglycerides (TG), or in derived forms, like SPMs, synaptamide and precursors.

Studies comparing the effect of supplementation with structured TGs and PLs ω-3 LCPUFAs against EE have proven enhanced bioavailability and powerful benefits of the structured forms in comparison with EE [90,91,92]. Similar intestinal absorption capacity and bioavailability between TG and PL have been described in studies analysing the responses of blood biomarkers [93,94], however, a greater brain absorption has been observed when DHA is esterified to PL [95]. In contrast, supplemented diets with a source of PL-DHA, TG-DHA or a mixture of both, resulted in similar increases in brain DHA compared to a low ω-3 PUFA diet [96].

Positional distribution is also important in structured lipids. For example, depending on the position of the fatty acid in the glycerol backbone of the TG (three positions, sn-1, sn-2 and sn-3), higher or lower absorption of the fatty acids can be observed. After intake, hydrolysis of TGs is performed by lipases, which are typically enantioselective. In the digestive tract, fatty acids in the sn-1 and sn-3 positions of the glycerol backbone are cleaved while the fatty acids of the sn-2 position are mainly absorbed [87] as represented in Figure 1. In this sense, preclinical results show differences in bioavailability when dietary TGs had DHA in the sn-1, sn-2 or sn-3 position, explained by less secretion of fecal DHA when this was at the sn-2 position. However, the same study shows no difference in DHA content of the fasting plasma, probably because the 5-day intervention in rats was not long enough to modify the fatty acid profile of phospholipids [97].

In addition to differences in the absorption, there are also divergences in the effect on the cells despite structured ω-3 LCPUFAs share common mechanisms with EE forms as schematically represented in Figure 2. In vitro studies using DHA-TG showed powerful antioxidant responses in comparison with EE-DHA in experiments with human fibroblasts and human retinal pigment epithelium cells through enhancing GSH synthesis significantly [17]. As mentioned before, some experimental studies have provided evidence about the interrelationship between DHA and GSH mainly through the regulation of nuclear factors like NFE2L2 [50,53], however TG-DHA seems to stimulate this reported pathway stronger than EE-DHA [17,48]. Moreover, a recent clinical trial proven significant anti-inflammatory properties after supplementation with TG-DHA through reduction of proinflammatory cytokines levels in patients with recurrent uveitis, one of the major causes of vision loss [88].

Concerning neuroinflammation and cognitive decline, in vitro and pre-clinical studies show improved efficacy when structured ω-3 LCPUFAs are used in comparison with EE [95] as detailed in Figure 2. One study where the effect of TG-DHA on microglial activation was assessed and compared with EE-DHA, proved that TG-DHA treatment protected microglia cells from oxidative stress toxicity by attenuating nitric oxide (NO) production and suppressing the induction of inflammatory cytokines [98]. Furthermore, in the same study, when 50 or 250 mg/kg of TG-DHA was given orally to mice with autoimmune encephalomyelitis for a total of 56 days, a significant amelioration of the course and severity of the disease as compared to untreated animals was observed, concluding that TG-DHA is a promising nutritional immunomodulating agent in neuroinflammatory processes [98]. Supporting this data, another study using a Parkinsonism murine model, which constitutes a powerful neurotoxic model, indicated that 250 mg/kg of TG-DHA for 22 consecutive days acted as a neuroprotective agent and may constitute a promising therapeutic adjuvant [99].

Looking for a greater bioactive and targeted effect, other types of structured ω-3 LCPUFA, in addition to PL and TG, have been tested in pre-clinical studies. This is the case with ω-3 LCPUFAs esterified with lysophosphatidyl-choline (LPC). When LPC is esterified with a saturated fatty acid it becomes a highly proinflammatory molecule, while esterification with a ω-3 PUFA, causes it to have the opposite role [100]. LPC has a great affinity for the receptor MFSD2A (sodium-dependent LPC symporter 1), a transmembrane transporter of long-chain fatty acids across the blood brain barrier (BBB) [101], located exclusively on the luminal membrane of endothelial cells that line the blood vessels in the brain [102]. Consequently, in vivo studies showed greater brain uptake of DHA and higher DHA enrichment of cell membranes in neural tissue when LPC-DHA was supplemented in comparison with PL and TG-DHA [103,104].

Another example of a special ω-3 LCPUFA is AceDoPC^®^ [105]. It is a structured DHA-PL acetylated at the sn-1 position—structurally similar to LPC-DHA—targeted to improve brain DHA levels. In an experimental ischemic stroke rat model, the intravenous injection of AceDoPC^®^ proved to have more powerful anti-inflammatory effects, attenuating induced neuroinflammation by decreasing IL-6 production [95], in addition to exerting more neuroprotective effects than DHA-EE in another study with a stroke rat model [106]. Moreover, a study with neural stem progenitor cells (NSPCs) derived from the adult mouse brain showed enhanced neurogenesis with AceDoPC^®^ over DHA-EE, especially under hypoxigenic (ischemic) conditions in vitro [107].

Furthermore, supplementation with synaptamide, which is mentioned as a bioactive form of DHA in the brain, has been assayed by Tyrtyshnaia et al. using synpatamide extracted from squid and administered subcutaneously to rats as a water emulsion. Synaptamide treatment attenuated microglial activation, release of proinflammatory cytokines, and decreased hippocampal neurogenesis in rats with a sciatic nerve chronic constriction injury [108].

Studies supplementing isolated forms of SPMs, derived lipid mediators of ω-3 LCPUFA, have been performed. Although limited literature relates the anti-inflammatory effect of SPMs with cognitive decline improvement [76], successful results have been observed with the use of supplements with SPMs and precursors of SPMs. It has been proved that supplementation with an oil enriched with SPMs and precursors, significantly increases SPM concentration in peripheral blood in humans [109], and larger intervention studies have been performed demonstrating that an orally administered SPM-enriched supplement improved the quality of life and reduced pain in a sample of adults with chronic pain [110].

Considering all the favourable literature, research to test the effect on brain health of new structures of ω-3 LCPUFA must go further to gain knowledge about the promising effects of targeting the positional distribution of ω-3 LCPUFA, and to find new chemical and enzymatic modification strategies to design special structures with targeted effects.

## 6. High-Throughput Techniques and Biological Models to Study Omega-3 LCPUFAs and Cognitive Decline

To discover the real potential of the ω-3 LCPUFAs on healthy aging not all experimental procedures can be entirely focused on nutritional interventions. Specific mechanisms must be elucidated, and fast screening methods for new structures of ω-3 LCPUFA are needed. The following points suggest high-throughput techniques and models that provide large information in a fast and cost-effective manner that might be very helpful in the study of ω-3 LCPUFAs.

### 6.1. “Omic” Technologies

Thanks to innovative breakthroughs in genome sequencing, bioinformatics, and analytic tools such as liquid (LC) and gas (GC) chromatography, mass spectrometry (MS), and nuclear magnetic resonance (NMR), “omics” technologies have appeared: genomics, transcriptomics, proteomics, metabolomics, metagenomics and epigenomics [111]. They are based on high-throughput identification and quantification of small and large molecules in cells, tissues, and biofluids, and are a powerful tool for mapping global biochemical changes and discovery of biomarkers [112]. Many omics disciplines are employed in food and nutrition research, and it is a prevalent recognition among food scientist that omics-based approaches are highly effective when they are exploited properly [113].

There are different motivations for conducting omic research, but commonly, they are performed to obtain a comprehensive understanding of the biological system under study, or to associate the omics-based molecular measurements with a clinical outcome of interest [114]. Researchers now put the combination of multiple omics analyses (integrated omics) into practice to exhaustively understand the functionality of food components of which nuclear NMR and MS are major choices. Generally, NMR is easier to perform and applicable to a wider range of compounds, although it is less sensitive compared to MS-based techniques. In contrast, GC or LC are used depending on the property of the target molecules [113].

The development of omic technologies brought about the expectation that an exhaustive molecular description of aging-regulated processes should have been possible, thereby shedding light on its mechanisms [115]. They would provide fast and precise information of specific and early biomarkers of the onset of homeostatic disturbances while aging, and this could help translational clinical research to describe quantitatively or qualitatively the health status of an individual or underlying aging mechanism [116,117]. For example, mass spectrometry-based omic technologies were used to reveal metabolic changes taking place during normal brain aging: metabolomic and proteomic analyses of different regions of mouse brain during the adult lifespan demonstrated an energy metabolic drift or significant imbalance in core metabolite levels in the aged animals [118].

Furthermore, omics technologies have already been applied in studies searching for the potential metabolic pathways and integrative biomarkers that would help to understand the link between nutrition, food patterns [119,120] and brain health [121]. A recent preclinical study with mice, highlighted the roles of neuroinflammation induced by gut dysbiosis and lipid metabolism disorders in Alzheimer’s progression, through an integrated metabolomic approach, showing the potential of omics techniques to understand complex mechanisms [122]. Also, in relation to the gut brain axis, another study found that a 28-day intermittent fasting regimen improved cognitive deficits in diabetic mice. They detected the microbiota-metabolites-brain axis alterations by multiple-omics analyses (transcriptomics, 16S rRNA sequencing and metabolomics), and the multi-omics approach found a correlation among gut microbiota, plasma metabolites, and hippocampal gene expression [123].

However, there are few articles with multi-omics analyses to study a close effect of ω-3 LC PUFAs or ω-3 PUFAs in preventing cognitive decline. Chakraborty et al. showed through the transcriptomic analysis of rodent hemibrain that a diet high in ω-3 PUFA stimulates the PI3K-AKT-PKC network, thus enhancing neuritogenesis (formation of new neurites) and reinforcing synapses [124], demonstrating a significant impact of a fish oil-enriched diet on nervous system development and neurological diseases. On the other hand, Kaliannan et al. using transgenic rodents and a multi-omics technologies approach, uncovered a potential pathway for the development of modern chronic diseases and cancer from the dietary imbalance between ω-6 and ω-3 PUFA. In this case, ω-3 PUFAs were detected as a biomarker for a potential risk factor [125].

In humans, the omic technologies are very focused on aging biomarkers detection, and already some reviews about promising biomarkers of human aging [116] or metabolomics of brain aging [126] can be found. Recent, literature examined whether ω-3 PUFAs, Vitamin D, and homocysteine formed into a nutritional risk index can explain cognitive performance of older non-demented adults. The study showed that executive skills were superior in those older participants with plasma EPA + DHA wt% ≥ 2.53, vitamin D (25-OH-D) ≥ 25 ng/mL, and homocysteine < 11.57 µmol/L [127].

As we are focused on the effect of ω-3 LCPUFA, lipidomic profiling, which is the metabolomic study of lipids in the tissues, cells and biological fluids, is essential. A recent study comparing the plasma lipidomic responses of 30-day supplementations of krill-oil versus fish-oil in healthy young women reported significant differences in the responses between supplements: krill-oil had a more pronounced effect on ω-3-containing PL species, in contrast to the fish-oil, which had a more significant effect on ω-3-containing neutral lipid species [128].

The most relevant study in relation to lipidomic profiling and cognitive decline, published by Mapstone et al. describes a lipidomic approach for detecting preclinical Alzheimer’s disease in a group of cognitively normal older adults. It was reported that a validated set of ten lipids from peripheral blood predicted short-term conversion to mild cognitive impairment or Alzheimer’s disease within a 2–3 year timeframe with over 90% accuracy [129]. These lipids have essential structural and functional roles in cell membranes, suggesting that their peripheral blood levels and lipid profiles could be used as an early multivariate biomarker with a well correlated ratio of neurodegeneration. Moreover, this specific lipid profile could be used as inclusion criteria in more well-designed human clinical trials of nutritional interventions that assess lipid effects on the prevention of cognitive decline [121].

Remarkably, novel concepts about the relation between the gut-brain axis and cognitive decline have been established through the application of metagenomics, since ω-3 LCPUFAs and its derived mediators have already proven the capacity to change the gut microbiota through their anti-inflammatory effects in preclinical models [130,131]. In humans, recent studies determined a correlation between ω-3 LCPUFAs and the gut microbiota thanks to their neuroinflammatory resolving effects [132]. Menni et al. found that the DHA intake of 350 mg/day was positively correlated with friendly bacteria through the use of next-generation DNA sequencing (*Lachnospiraceae family*) in middle aged and elderly women [133]. Moreover, a multicentre clinical trial with patients diagnosed with type 2 diabetes showed a change in the gut microbiota with a decrease in the Firmicutes/Bacteroidetes ratio and an increase in the *Prevotella genus* after a sardine-enriched diet [134]. Altogether, the supporting literature make it clear that multi-omic approaches are essential to study the relation between brain aging and ω-3 LCPUFAs.

### 6.2. Fast and Cost-Effective Experimental Models

The place of animals in our modern societies, especially of mammals, is often debated, particularly the right to use mammals to benefit human purposes when there is the possibility that they will be harmed. Moreover, not all results obtained from animals, mainly rodents, can be directly translated to humans, although there are remarkable anatomical and physiological similarities [135]. Consequently, more researchers have started to use alternative in vivo models, such as invertebrates or fish, rather than mammals which are between in vitro and rodent models, to obtain information from a complete organism in a fast and cost-effective manner. They are especially useful for the study of regulatory pathways and cellular mechanisms, as well as being suitable as screening platforms to test drugs and bioactive compounds. In addition, since the food industry is getting more and more involved in health issues by designing innovative foods that contribute to a better nutritional profile or to a certain functionality, fast and cost-effective models can be used to pre-screen compounds with bioactivity to speed up the demonstration of active ingredient effectiveness [136]. As represented in Figure 3, the simultaneous combination of these sorts of biological models with a biology systems approach using omic technologies, will make it easier and more feasible to scale-up the business pipeline of a bioactive ingredient until scientific and legal requirements, in terms of demonstrating efficacy are met.

#### 6.2.1. *Caenorhabditis elegans* (*C. elegans*)

The nematode *C. elegans*, frequently used to study aging because of its lifespan characteristics, has emerged as a powerful model organism for drug or nutrient screening due to its cellular simplicity, genetic amenability and homology to humans combined with its small size and low cost [137].

Morphologically, the adult *C. elegans* hermaphrodite form has only 959 somatic cells, but it has various organs and tissues, including a nervous system consisting of only 300 neurons and 56 glial cells. The genome of the nematode shares greater than 83% homology with the human genome, sharing human disease genes and disease pathways. In addition, the main functional components of mammalian synaptic transmission, such as neurotransmitters, ion channels, receptors, and transporters, are conserved in *C. elegans* [138].

Many studies have demonstrated the usefulness of *C. elegans* as a tool to study the underlying mechanisms that give rise to aging-associated neurodegenerative diseases, including Alzheimer’s disease [139,140], and other age-related conditions like sarcopenia, because this nematode also shows loss of muscle mass with aging [141]. Moreover, *C. elegans* not only allows effects in behavioural and molecular improvements to be determined, but also modulation of lifespan caused by the studied substance [142].

However, *C. elegans* has a different lipid metabolism in comparison to mammals [143]. Unlike most other animal species, the *C. elegans* genome encodes an ω-3 desaturase enzyme that can convert 18-carbon and 20-carbon ω-6 fatty acids into ω-3 fatty acids, along with a ∆12 desaturase, which catalyses the formation of linoleic acid from oleic acid. Thus, *C. elegans* does not have any dietary fatty acid requirements. Like most animals, *C. elegans* also possesses ∆6 and ∆5 desaturase enzymes, which act, in conjunction with fatty acid elongases, on similar substrates used by mammals and other animals to form 20-carbon PUFAs. By contrast, the nematode is not able to produce 22-carbon PUFAs due to a lack of the specific elongase activity. These differences in the LCPUFAs’ metabolism, in combination with the simple anatomy of *C. elegans* and a range of available genetic tools, makes this organism an attractive model to study fatty acid function. In this sense, strains containing mutations in genes of the fatty acid desaturation pathway facilitate functional studies of PUFAs, and fatty acid composition can be manipulated both genetically and through the diet [144].

There are already many bioactive compounds, especially polyphenols and flavonoids derived from plant extracts [145], which have successfully demonstrated an anti-aging effect in *C. elegans* and curiously through a similar mechanism [146]. For example, trials with curcumin [147], *Hibiscus sabdarifa* extract [148], strawberry extract [140] and *Ilex paraguariensis* extract [149] agree in regard to the activation of the transcription factors DAF-16 and/or SKN-1 in *C. elegans*, which have a homologous function to FOXO and Nrf2 (encoded by NFE2L2) in humans, respectively [150]. In mammals, the FOXO factor activates the expression of genes related to longevity by promoting, among others, the synthesis of sirtuins (histone deacetylase enzymes), and the factor Nrf2 is responsible for inducing the expression of genes encoding the enzymes of the antioxidant response.

Nevertheless, there is still few research studies assessing the effect of ω-3 PUFA using *C.elegans* as an experimental model. According to a study published some years ago, ω-3 PUFAs are able to extend *C. elegans* lifespan through PPARα activation [151]. In contrast, assessment of fish oil in *C. elegans* gave opposite results, with ω-3 PUFAs also being observed in high doses of oil to cause a reduction in lifespan. This is probably because excessive amounts of ω-3 PUFAs are more likely to undergo peroxidation, thus increasing ROS production and ultimately leading to a lifespan reduction. This suggest that low concentrations of ω-3 PUFAs are better able to improve lifespan [141], a phenomenon that makes sense with the hormetic effect of ω-3 LCPUFAs, since it has already been reported that low doses of phytochemicals [152] and fatty acids [153] emphasize stress-response pathways, enhancing the regulation of various cytoprotective proteins.

*C. elegans* has also been used as a model to test different lipidic structures. Synthetically structured TGs with medium and long-chain fatty acids were administered in an emulsion to the nematode. The results showed that medium and long-chain TGs could shorten the lifespan and increase ROS, despite TGs with mixed combinations of medium and long-chain fatty acids not affecting the lifespan of the nematodes [89]. Furthermore, Beaudoin-Chabot et al. performed a study where *C. elegans* was supplemented with deuterated trilinolenin, a TG with 3 α-linolenic acid (ALA, the precursor of the ω-3 PUFA series). Using a knockout strain of *C. elegans* (*fat-1*) that mimics the human dietary requirement of ω-3 fatty acids, they proved that the deuterated lipid significantly extended the lifespan of worms, prevented the accumulation of lipid peroxides and reduced the accumulation of ROS, demonstrating that deuterated ALA-TG could be used as a food supplement to decelerate the aging process [154].

In addition, the nematode is a suitable model for multi-omics approaches. For example, metabolic changes that characterize wild type and long-lived *C. elegans* strains have been previously studied by applying metabolomics techniques [155]. However, a common challenge related to omics applied to aging studies is that *C. elegans*, as a hermaphroditic species, can contaminate aging populations with progeny introducing a confounder factor. The use of 5′-fluoro-2′-deoxyuridine (FUDR) can circumvent this problem because it decreases progeny production by reducing germ cell division, but this treatment has been shown to directly influence metabolism. To solve this problem, comprehensive time-resolved multi-omics and modelling resource for studying the metabolic changes during normal aging in *C. elegans* have started to be developed [156]. Furthermore, several microfluidic systems for *C. elegans* have been developed recently, allowing automation and performing aging experiments in a high-throughput manner [157].

To sum up, despite *C. elegans* possibly being sensitive to treatments with ω-3 PUFAs, it is a great model to study the mechanisms involved in the relationships between ω-3 PUFAs and aging [144]. It is faster and cheaper than other in-vivo models, such as murine models, and it is very versatile because of its lipid metabolism, so it could be very useful for screening assays of structured ω-3 LCPUFAs.

#### 6.2.2. *Drosophila melanogaster* (*Drosophila*)

*Drosophila* is a species of fly often referred to as the fruit fly. It is typically used in research owing to its rapid life cycle, relatively simple genetics with only four pairs of chromosomes, and large number of offspring per generation [158]. Despite *Drosophila* being predominantly used as a model to understand developmental biology, the flies have also been used for testing new drugs in a much faster way than mammalian models; indeed they may even be used for the initial high-throughput screening process as an alternative to cell culture [158]. In comparison with *C. elegans*, drug administration can be more complicated because *Drosophila* is not so close to the medium.

Furthermore, with a median lifespan of about 60–80 days and well-conserved metabolic pathways between fly and man, *Drosophila* has emerged as an excellent genetic model to study the complexity of the aging process. Added to the ability to efficiently identify and characterize single-gene mutations that extend lifespan, the well-developed genetic techniques that allow precise spatiotemporal control of genetic perturbations have made flies the premier model system to address questions about tissue-specific functional decline and tissue–tissue interactions during the aging process [159].

Curiously, diverse studies with ω-3 LCPUFAs have been made in *Drosophila* and DHA and EPA have already proven to significantly increase longevity. The fly is an interesting model to understand the fundamental mechanisms that control lipid metabolism because it does not possess the ∆5 and ∆6 desaturases which participate in the conversion of ALA to EPA and to DHA [160]. Champigny et al. tested monoglyceride forms of EPA and DHA in *Drosophila* that showed increased bioavailability in previous studies using rodents. A dose of 0.3 mg/mL of these monoglycerides showed potent effects on mitochondrial respiration, specially EPA, and provided protection against lipid peroxidation by maintaining SOD activity during aging, demonstrating that *Drosophila* can provide a new understanding of the potential beneficial effects of investigating the metabolic pathways of DHA and EPA in oxidative stress related pathologies [160].

In addition, ω-3 LCPUFAs showed neuroprotective effects in *Drosophila*. Their ingestion protected the flies against oxidative stress and consequent neuronal and mitochondrial dysfunctions frequently found in neurodegenerative processes, reinforcing the protective role of EPA and DHA against environmental neurodegenerative diseases [161]. Moreover, the proven neuroprotective effect has been linked to improvements in cognitive function in *Drosophila*. For example, the supplementation of fly dietary pattern with fish oil positively impacts short-term memory and learning abilities compared to fruit flies that are provided with a standard diet [162].

As a high-throughput screening platform, omic technologies applied to the aging process have also been used in *Drosophila*, and for example, it has been measured how the global lipidome of *Drosophila* changes during aging. In this study selective degradation of TG species in flies close to death were found while PL signatures were almost unaltered compared to normal flies at all ages, suggesting a tight control of membrane composition throughout lifetime largely uncoupled from storage lipid metabolism [163].

#### 6.2.3. Zebrafish

The zebrafish (*Danio rerio*) is a freshwater fish, traditionally used in ecotoxicology and developmental biology studies. As an experimental organism, it has expanded to other fields like regenerative medicine, infectious diseases, neurosciences and cancer research among others [164]. Compelling experimental features together with its similarity with mammals are key factors for its popularity. Zebrafish share organ make up, cellular types and metabolic processes with vertebrates, and have orthologous genes for 70% of human ones. However, it is mainly due to its ease of use that it is such a powerful research tool. The zebrafish life cycle is short, and they are highly fecund, spawning hundreds of offspring per breeding couple per week. All these advantageous features have made the zebrafish a valuable model to demonstrate bioactivity in food research and it has been recently introduced into this field [136].

As a model used for toxicological studies, zebrafish have been extensively used to screen molecules with antioxidant or pro-oxidant potential by different methodologies, and it is a suitable model to determine the pro-inflammatory resolving properties of bioactive compounds [136]. A recent study testing aspirin in zebrafish, linked neuroinflammation with cognitive defects showing decreased neutrophil infiltration and changes in the neurons of the hippocampus in sleep deprived fish [165]. Despite limited literature regarding this phenomenon in zebrafish, new data reinforce the growing utility of zebrafish to explore neurobehavioral bases and provide clinically translatable data [166]. Cognitive function can be easily measured in zebrafish using learning tasks and short-term and long-term memory tests, and neurodegenerative disorders can be assessed measuring decline in locomotion and characteristic biomolecular and cellular markers [164].

Altogether, there is also extensive literature testing bioactive compounds and nutritional interventions in zebrafish [167]. For example, flavonoids have demonstrated chemobehavioral effects in zebrafish larvae [168], and fish fed with high-fat diets had worse performance in behavioural tests [166].

Zebrafish are an interesting model organism for lipid metabolism studies because they synthesize LCPUFA through a similar pathway to the one used by humans, despite not expressing FADS1 like mammals [169]. As an example, Sierra et al. showed that DHA had antiepileptic effects in a zebrafish model [170] and more recently, a study supplementing ALA proved a reduction in epileptic seizure susceptibility in developing zebrafish embryos [171]. However, there are still few studies about prevention of cognitive decline and ω-3 LCPUFAs in fishes.

New omics technologies have also been used with zebrafish in food research. As examples, nutrigenomic analysis was performed in zebrafish after consumption of poultry egg products [172], and metagenomic analyses were made after a 6-month feeding trial with *Hermetia illucens* [173].

In summary, despite zebrafish having considerably longer life cycles in compared to worms and flies, fish husbandry is considerably more complex because animals must be maintained in centralized aquaculture facilities with controlled water and light parameters. Despite this zebrafish are a good model for ingredient suppliers and scientists to test the impact on health of several compounds, particularly, because they have a high conservation of physiological processes and genes relevant to human disease [174].

## 7. Conclusions

Nutritional strategies have been widely applied in the preservation of brain health since food contains a variety of bioactive substances closely related with proper cognitive function. Vitamins, minerals, polyphenols and ω-3 LCPUFAs are key nutrients for healthy brain aging which reduce age-related inflammation and oxidative stress, according to preclinical data. Considering the role of DHA and EPA, they have promising potential against mental health impairments, but for the time being, the relationship between cognitive function and individual food components remains inconclusive because no clear efficacy or mechanism has been confirmed in humans.

Recently, new structured forms of both DHA and EPA with improved bioavailability gained attention as they could exert a greater effect in reducing neuroinflammation and enhancing cognitive functions. This review summarizes the current state of the art in relation to these molecules and suggests a research pipeline based on the use of high-throughput in vivo models as screening platforms i.e., zebrafish, *C. elegans* and *Drosophila,* in combination with cutting-edge omic technologies. The use of these innovative tools and strategies will provide a reliable set of data to comprehend and shed light on the effect and mechanisms of new structures of ω-3 LCPUFAs on cognitive functions and brain aging in a cost-effective manner that could be applied in the food industry (Figure 3).

In closing, the promising potential of new structural forms of DHA and EPA as bioactive compound for healthy brain aging could be used in more personalized dietary patterns and in the maintenance of optimum brain health in elderly people, improving healthspan and contributing to a more sustainable healthcare system.

## Figures and Tables

**Figure 1 ijms-23-03472-f001:**
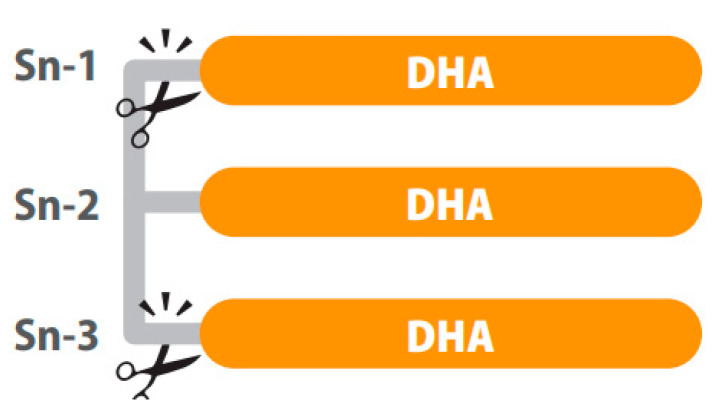
A triglyceride of DHA digested by lipases. The sn-1 and sn-3 positions of the glycerol backbone are cleaved while the fatty acids of the sn-2 position remains. Image courtesy of Brudylab^®^.

**Figure 2 ijms-23-03472-f002:**
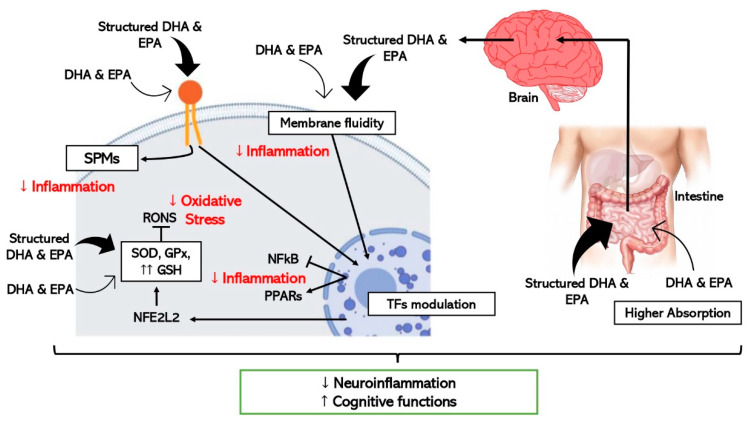
Schematic representation of the hypothetic mechanisms through which ω-3 LCPUFAs reduce neuroinflammation and improve cognitive function by modulating inflammation and oxidative stress in brain cells. Thick arrows express the enhanced effect of the structured forms of DHA and EPA (TFs, transcription factors).

**Figure 3 ijms-23-03472-f003:**
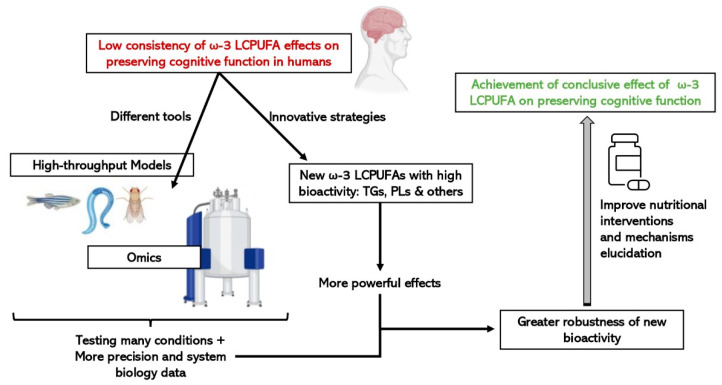
Advantages of integrating the use of omic technologies and small high-throughput in vivo models on the study of the efficacy of new structures of ω-3 LCPUFAs on cognitive function.

## Data Availability

Not applicable.

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
