# Peer review of "Structured Long-Chain Omega-3 Fatty Acids for Improvement of Cognitive Function during Aging"

_ijms, 2022, doi:10.3390/ijms23073472_

Round 1
Reviewer 1 Report
The review article “Structured Long-Chain Omega-3 Fatty Acids for Improvement of Cognitive Function During Aging” is dedicated to summarizes present and future perspectives of new structures of ω-3 LC-PUFAs and the role of omic technologies and high-throughput in vivo models to shed light on the relation and underlying mechanisms between ω-3 LC-PUFAs and healthy aging.
The article is well written.
The study has a good design.
The article is logically divided into sections and subsections.
In the article there are no grammatical and stylistic errors.
There are many figures of good quality presented in the article.
The references cited are relevant and adequate.
A large number of scientific literature sources were analyzed.
The work has a high degree of novelty.
In my opinion, this review paper can be recommended for publication after minor revision.
It is recommended to expand section “Antioxidant pathway.”
It is recommended to include a list of abbreviations, used in the article.
Author Response
First, thank you for your positive feedback, I am glad you liked the review.
Changes:
- The section “Antioxidant pathway” has been expanded. The pathways through DHA promotes the antioxidant response are now more cohesive and detailed. Also new references have been added.
- A list of abbreviations have been included.
Please see the attachment to read the new sections new section.

Reviewer 2 Report
This manuscript reviews the current and future perspectives of new structures of ω-3 LC-PUFAs and the role of omic technologies and high-throughput in vivo models to shed light on the relation and underlying mechanisms between ω-3 LC-PUFAs and healthy aging. This is a good review article with a lot of information for the relevant researchers or interested readers. The reviewer has only two minor comments that the scientific names in the headings on lines 525 and 526 should still be in italics and the authors should provide structural diagrams of the important structured long-chain omega-3 fatty acids mentioned in this manuscript.
Author Response
First, thank you for your positive feedback. We are glad you liked the review.
Changes:
- The headings on lines 525 and 526 are now in italics
- I added a structural diagram of a structured form of DHA proving the relevance of the positional distribution of the long-chain omega-3 fatty acids. Please see the attachment.
Thank you for your time.
